# Highly diverse–Low abundance methanogenic communities in hypersaline microbial mats of Guerrero Negro B.C.S., assessed through microcosm experiments

**Patricia J. Ramírez-Arenas**[1], **Hever Latisnere-Barragán**[1], **José Q. García-Maldonado**[2]*, **Alejandro López-Cortés**[1]*

**1** Centro de Investigaciones Biológicas del Noroeste (CIBNOR), La Paz, Baja California Sur, México,
**2** Centro de Investigación y de Estudios Avanzados del Instituto Politécnico Nacional, Unidad Mérida, Mérida, Yucatán, México

\* jose.garcia@cinvestav.mx (JQGM); alopez04@cibnor.mx (ALC)

## Abstract

Methanogenic communities of hypersaline microbial mats of Guerrero Negro, Baja California Sur, Mexico, have been recognized to be dominated by methylotrophic methanogens. However, recent studies of environmental samples have evidenced the presence of hydrogenotrophic and methyl-reducing methanogenic members, although at low relative abundances. Physical and geochemical conditions that stimulate the development of these groups in hypersaline environments, remains elusive. Thus, in this study the taxonomic diversity of methanogenic archaea of two sites of Exportadora de Sal S.A was assessed by *mcrA* gene high throughput sequencing from microcosm experiments with different substrates (both competitive and non-competitive). Results confirmed the dominance of the order Methanosarcinales in all treatments, but an increase in the abundance of Methanomassiliicoccales was also observed, mainly in the treatment without substrate addition. Moreover, incubations supplemented with hydrogen and carbon dioxide, as well as the mixture of hydrogen, carbon dioxide and trimethylamine, managed to stimulate the richness and abundance of other than Methanosarcinales methanogenic archaea. Several OTUs that were not assigned to known methanogens resulted phylogenetically distributed into at least nine orders. Environmental samples revealed a wide diversity of methanogenic archaea of low relative abundance that had not been previously reported for this environment, suggesting that the importance and diversity of methanogens in hypersaline ecosystems may have been overlooked. This work also provided insights into how different taxonomic groups responded to the evaluated incubation conditions.

## Introduction

Recent records of the rising concentrations of atmospheric methane have driven the interest to understand the sources of this important greenhouse gas [1] to accurately predict future trends

**Data Availability Statement:** Data have been deposited into NCBI under BioProject accession

**Funding:** This work was supported by Consejo Nacional de Ciencia y Tecnología (CONACYT) through grant FORDECYT-PRONACES, CF-2019-848287 to A.L.-C. and J.Q.G.-M. The funding institution, Consejo Nacional de Ciencia y Tecnología (CONACYT), had no role in study design, data collection and analysis, decision to publish, or preparation of the manuscript.

**Competing interests:** The authors have declared that no competing interests exist.

of global warming [2]. More than half of global methane emissions are derived from microbial activity [3], and the archaeal-specific metabolism is of key relevance in the anaerobic degradation of organic matter and biogas production [4]. It is stemming mostly from natural wetlands and sediments but is also common in environments presenting extreme temperatures, salinity, and pH [3]. It should be noted that most organic substances, for instance, carbohydrates, long-chain fatty acids and secondary alcohols, are not direct substrates for methanogenesis. Instead, these compounds must first be processed by intermediary microorganisms as bacteria or unicellular eukaryotes to produce hydrogen plus carbon dioxide, acetate, or methylated compounds, which are the substrates that methanogens would actually use [5]. Four pathways of anaerobic methanogenesis have been recognized to date: hydrogenotrophic (using $H_2/CO_2$, formate, and CO as substrates), acetoclastic (acetate), methylotrophic ($C_1$-methylated compounds), and methyl-reducing ($C_1$-methylated compounds as acceptors plus $H_2$ or formate as donors) [6]. In hypersalinity conditions such as those that characterize the microbial mats of the Exportadora de Sal S.A. saltern in Guerrero Negro, Baja California Sur, Mexico, methanogenesis is controlled by a gradient of redox potential and the high concentration of sulfate [7]. This is because sulfate-reducing bacteria (SRB), compared with methanogens, have a greater affinity and energy yield, when competitive substrates like formate, hydrogen, and acetate, are available [8]. However, high concentrations of noncompetitive substrates (e.g., methanol, monomethylamine, dimethylamine, trimethylamine (TMA), and dimethylsulfide) derived from organic osmolytes such as glycine betaine or dimethylsulfoniopropionate, drive methane production in these ecosystems [7].

Several studies have addressed the diversity, abundance, and physiology of methanogens from hypersaline environments [9–15], indicating the predominance of methylotrophic methanogens of the order Methanosarcinales. Nonetheless, a recent survey [16] based on 16S rRNA and methyl coenzyme M reductase (*mcrA*) gene sequences of hypersaline microbial mats revealed lineages of methanogens whose taxonomic assignment relates them to other type of metabolisms, suggesting that methane production in this environment may be underestimated and hence, not be limited to a single methanogenic pathway. Considering this, the present work aimed to assess the stimulation of low abundance methanogenic groups by substrates and electron donor additions; and to elucidate the changes in methanogenic communities when these conditions are present in the environment. The results provided insights into the low-abundance methanogenic diversity, phylogeny and its potential contribution to ecosystem functioning in the hypersaline environments of Guerrero Negro.

## Materials and methods

### Sampling site and field measurements

Portions of well-consolidated and well-laminated, submerged microbial mats (20 cm x 30 cm) were sampled in April 2022 from two brine evaporation ponds: Area 4 near Area 5 (A4N5) (27.687N, -113.917W) and Area 5 (A5) (27.690N, -113.917W) located at Exportadora de Sal S. A., Guerrero Negro, Baja California Sur, Mexico (Fig 1), that kindly provided access to their facilities and allowed us the collection of samples. The salinity, temperature, and pH of the overlying water were determined in situ with a portable salinity refractometer (Vista, China), digital thermometer (VWR international), and pH meter (Orion, Beverly, MA, USA), respectively.

For molecular analyses, the mat portions collected were kept immersed in the pond brine from each site, stored with ice and transported to the laboratory. A total of five microbial mat cores (1.0 cm diameter x 0.5 cm deep) were sub-sampled as described previously [16] and preserved in 5 ml of RNAlater (Thermo Fisher, Carlsbad, CA, USA), contained in 15 ml tubes.

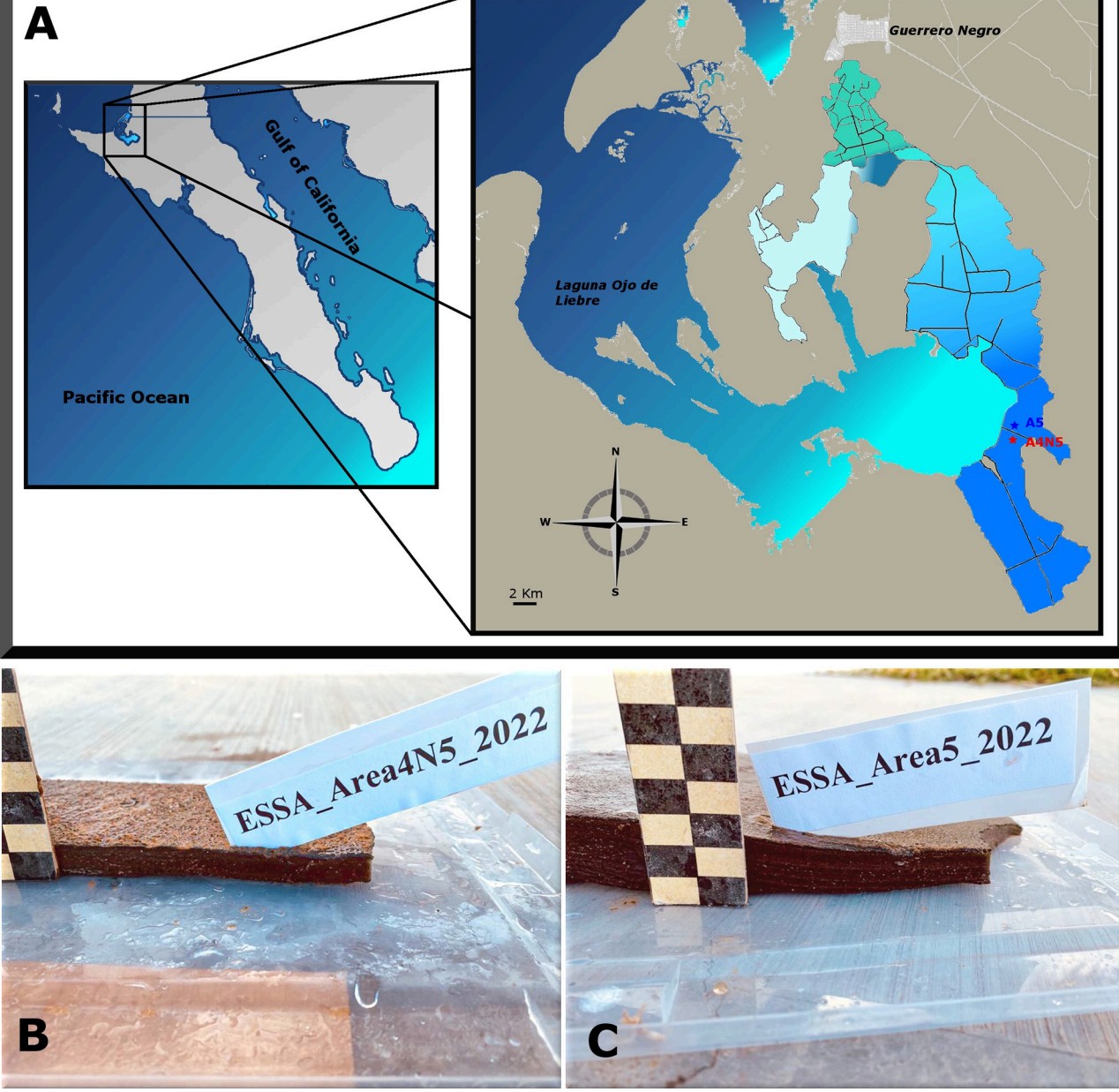

**Fig 1. Geographic location of sampling sites in Baja California Sur, Mexico.** A) Sampling site locations are indicated by red (A4N5) and blue (A5) position markers. B) Microbial mat samples collected in April 2022. Each square corresponds to 1 cm. Maps Source: INEGI, Digital topographic map, 2024, México (https://gaia.inegi.org.mx/mdm6). The original maps were modified to show the study sites. Map modifications comply with terms of use required by INEGI for free distribution (https://www.inegi.org.mx/contenidos/inegi/doc/terminos_info.pdf).

These samples (hereafter named "environmental samples") were not submitted to any treatment. Additionally, seven mat cores were stored immersed in artificial brine (8% salinity, 73 mM sulfate, pH 8.1), formulated with analytical grade chemicals (MERCK, KGaA, Darmstadt, Germany) as previously described [17]. The salinity and sulfate concentrations were settled according to a chemical analysis performed on samples of natural brines during a 2019 fieldwork (74.53mM). The immersed mat cores were kept at room temperature and dark conditions until the microcosm experiments were assembled.

## Microcosm experiments

To assess the effect of methanogenic substrates addition in the microbial community structure, three different treatments were carried out by triplicate: a) $H_2/CO_2$ atmosphere (80–20%; chromatographic grade 99.995%); b) $H_2/CO_2$ atmosphere (80–20%) + trimethylamine (15 mM; ACROS organics); and c) $N_2$ atmosphere (chromatographic grade 99.997%) + trimethylamine (15 mM; ACROS organics). The selected methanogenic substrates were intended to stimulate ubiquitous, highly diverse, low-abundance microbes, with potential hydrogenotrophic ($H_2+CO_2$) and methyl reductive ($H_2+CO_2+TMA$) methanogenesis metabolisms. In turn, the use of TMA served as a positive control for methylotrophic methanogenesis, with $N_2$ employed to replace the oxic atmosphere in the microcosm experiments. Additional triplicated incubations, without methanogenic substrates were performed as controls. For the experiment, the seven cores (~ 15 g) of hypersaline microbial mat samples were incubated with 20 ml of the artificial brine in sterilized serum vials capped with butyl rubber stoppers and aluminum crimps. After the addition of the cores to the artificial brine containing vials, they were purged during 30 s with escaping $N_2$ gas to displace the oxygen. According to the type of experiment, $N_2$ or $H_2/CO_2$ gas was finally injected during two minutes to the hermetically closed vials. All treatments were incubated in darkness at 22°C through 85 days.

## DNA extraction

After the 85 days of the experiment elapsed, total genomic DNA was extracted from one core of each replicate (~ 2.0 g), from all the treatments and from the environmental samples using the QIAGEN "DNeasy PowerBiofilm" kit (GmbH, Hilden, Germany) with some modifications. The manipulation of the mat cores was handled under sterile conditions to avoid any cross contamination. To discard all the inorganic material, each mat core was submerged in 2 ml of sterile PBS 4X and mechanically macerated with sterile polypropylene pestles and an electrical homogenizer (VWR, Wayne, PA, USA). The organic phase was recovered and centrifuged per 5 min to 10,000 rpm. A total of 0.150 g of the organic slurry, as well as a volume of 150 μl of the supernatant, which contained an additional organic phase that did not sediment with the centrifugation step, were recovered and processed following the kit's instructions. A negative control (column without sample) was processed together with the target samples to rule out possible cross-contamination of the kit reagents. The DNA integrity and concentration were assessed with standard agarose gel electrophoresis and spectrophotometric reads (NanoDrop Technologies, Thermo Scientific. Wilmington, DE, USA).

## Library prep and Illumina mcrA gene sequencing

Owed to the samples nature and in order to determine the efficiency of the PCR reactions, all the *mcrA* gene assays (25 μL) were initially performed with primers mlas-mod-F and mcrA-rev-R containing Illumina adapters [16], following the next conditions: 6.5 μL sterile water, 2.5 μL of each primer solution (10 μM), 12.5 μL GoTaq master mix (Promega, Madison, WI, USA) and 1 μL (10 ng μL$^{-1}$) DNA. The thermocycling conditions were performed in a Thermocycler Touch C-1000 (Bio-Rad, Berkeley, CA, USA) and consisted in one cycle of 95°C for 5 min, five touchdown cycles of 95°C for 30 s, 60°C for 45 s, diminishing 1°C per cycle and an elongation step of 72°C for 1 min, followed by 35 cycles of 95°C for 30 s, 54.5°C for 30 s, 72°C for 1 min and a final elongation step at 72°C for 5 min. Differences in the amplification efficiency were observed for both environmental and microcosm samples. Under the thermocycling conditions mentioned above, all the microcosm samples amplified a sharp but faint PCR fragment which required performing technical replicates to gather sufficient genetic material for the metagenomic sequencing process. Regarding the environmental samples, it was

necessary to perform a pseudo-nested assay in two steps. A first PCR assay with reduced thermocycling conditions was performed with the primers mlas-mod-F and mcrA-rev-R without the Illumina adapters, as follow: one cycle of 95˚C for 5 min, five touchdown cycles of 95˚C for 30 s, 60˚C for 45 s, diminishing 1˚C per cycle and an elongation step of 72˚C for 1 min, followed by 10 cycles of 95˚C for 30 s, 55˚C for 30 s, 72˚C for 1 min and a final elongation step at 72˚C for 5 min. The resulting PCR products were used as a template for a second PCR assay per triplicate with the primers mlas-mod-F and mcrA-rev-R containing the Illumina adapters under the same reaction conditions. For this second PCR assay, the thermocycling conditions were as follows: one cycle 95˚C for 3 min, 30 cycles of 95˚C for 30 s, 55˚C for 45 s, 72˚C for 1 min and a final elongation step of 72˚C for 5 min.

All the PCR products were purified using AMPure XP magnetic beads (Beckman Coulter Genomics, Brea, CA, USA) and indexed using the Nextera XT Index Kit (Illumina, San Diego, CA, USA) according to the Illumina Metagenomic Sequencing Library Preparation Manual. Barcoded PCRs were repurified as described above and then quantified using a Qubit 3.0 fluorometer (Life Technologies, Malaysia). The correct size of the libraries was verified on a QIAxcel Advanced system (GmbH, Hilden, Germany). Paired-end sequencing (2 × 250 bp) was performed at CINVESTAV Mérida using the MiSeq platform (Illumina, San Diego, CA, USA) with a 500-cycle MiSeq Reagent Nano Kit v. 2. The raw sequencing data generated in this study have been deposited at NCBI under the PRJNA1086882 BioProject accession number.

## Sequence data bioinformatic analyses

**Sequence quality control and taxonomic annotation.**   Quality control of paired *mcrA* gene sequences was performed using FASTP ver. 0.23.1 [18] with default parameters. High-quality trimmed reads were merged using Flash2 ver. 2.2.00 [19] with a minimum overlap length of three nucleotides. Amplicons shorter than 320 bp were discarded and for taxonomic assignment, the merged amplicons (non-chimeric and high quality) were translated and blasted against a local database built using all available entries for EC 2.8.4.1 from UniProt (UniProt Consortium 2022) (38 SwissProt and 6755 TrEMBL sequences, downloaded on July 14, 2023). Alignment parameters were set to report the top five hits. Operational Taxonomic Units (OTUs) were generated based on sequence similarity (97% identity) and alignment BLASTx score [20]. Chimeric sequences were removed employing Usearch v. 10.0.240 [21] and sequences with no hits were discarded from further analysis.

Taxonomic labels of OTU representative sequences were retrieved from the BLASTx best hit against the UniProt database described. Taxonomic rank names were obtained using the NCBI taxonomy database via TaxonKit v.0.10.1 [22].

**Diversity and statistical analysis.**   The OTUs abundance table was exported to the R environment [23] for statistical analysis and visualization using the microeco package v1.7.1 [24] and the vegan [25], phyloseq [26], and ggplot2 [27] libraries. To assess differences in alpha diversity among the sampling sites and between environmental replicates and methanogenic substrate incubations, mean values were used for statistical analysis using the nonparametric Kruskal-Wallis test. The similarity patterns of the beta diversity of microbial communities were analyzed using a non-metric multidimensional scaling (NMDS) approach, with Bray-Curtis distance matrices employed as the basis for the analysis.

**Phylogenetic analysis.**   OTU sequences without taxonomic assignment were translated into their corresponding Mcra peptide sequences via EMBOSS Transeq [28] tool. Maximum likelihood phylogeny was calculated using Pplacer v1.1 [29], with a McrA protein alignment provided by Dr. Luke McKay (Montana State University, Department of Land Resources and Environmental Sciences) as reference. Once the multiple sequence alignment of query and

**Table 1. Sampling sites coordinates and physicochemical parameters at Exportadora de Sal, S.A., Guerrero Negro, BCS, México.**

| Site | Coordinates | Salinity (‰) | Temperature (˚C) | pH |
|---|---|---|---|---|
| Area 4 Near Area 5 (A4N5) | 27.687 N, -113.917 W | 80 ± 0.01 | 20.1 ± 0.10 | 8.11 ± 0.003 |
| Area 5 (A5) | 27.690 N, -113.917 W | 85 ± 0.50 | 21.5 ± 0.05 | 8.20 ± 0.001 |

Values correspond to the arithmetic mean of triplicate measurements of the interstitial water from microbial mat samples.

reference sequences was performed using MAFFT v.7.487 [30], unaligned portions at the ends of amino acid sequences were trimmed with Jalview v2.11.1.4 [31]. The phylogenetic tree obtained by Pplacer was exported to Newick format for online viewing and editing via iTOL v6 [32].

## Results

### Field sites characterization

The studied microbial mats were roughly six centimeters thick with soft, dense, and well-laminated structure (Fig 1). The physicochemical properties of the brine that were measured in this study are shown in Table 1.

### Methanogenic microbial diversity and species richness

In this study, a total of 525,713 raw reads were obtained from *mcrA* gene sequencing. After denoising and chimera removal, 271,932 (51.72%) high-quality sequences were recovered. Rarefaction curves of cumulative increase of OTUs richness (mean) indicated that the sequencing depth was sufficient to capture most of the methanogenic diversity in each sample (S1 Fig). The diversity metrics shown in Table 2, revealed significant differences in microbial diversity between A4N5 and A5 environmental samples and their cognate methanogenic substrate incubations (Kruskal-Wallis, A4N5 p = $7.328e^{-15}$; A5 p = $1.257e^{-4}$); indicating a greater diversity and richness of OTUs for environmental communities of both sites. The average methanogenic richness values for the microbial communities of A4N5 and A5 ranged from 2.66 to 37 OTUs and from 4.02 to 22.3 OTUs, respectively (Table 2). Shannon diversity index values were highest for the environmental samples, with no statistically significant differences (p > 0.05) between the two sampling site medians.

**Table 2. Methanogenic community descriptors of environmental samples and substrate enriched A4N5 and A5 microbial mat microcosm essays, revealed by high-throughput sequencing of *mcrA* gene.**

| Site | Treatment | Reads | Observed OTUs | Shannon |
|---|---|---|---|---|
| A4N5 | Environmental | 6979 ± 1195 | 37 ± 6.32 | 2.36 ± 0.55 |
| | Control | 11202 ± 4029.3 | 9 ± 1.81 | 0.77 ± 0.47 |
| | $H_2+CO_2$ | 11805 ± 2010.2 | 7 ± 2.31 | 0.75 ± 0.41 |
| | $H_2+CO_2+TMA$ | 7260± 1841.5 | 3.66 ± 2.33 | 0.36 ± 0.25 |
| | TMA | 6817 ± 2419.6 | 2.66 ± 1.52 | 0.17 ± 0.36 |
| A5 | Environmental | 9295 ± 1792.1 | 22.3 ± 5.73 | 1.78 ± 0.43 |
| | Control | 7553 ± 432.5 | 4.02 ± 1.35 | 0.83 ± 0.37 |
| | $H_2+CO_2$ | 6573 ± 1404.6 | 5.58 ± 1.5 | 0.51 ± 0.78 |
| | $H_2+CO_2+TMA$ | 11802 ± 7353 | 6.46 ± 2.88 | 0.72 ± 0.48 |
| | TMA | 12648 ±4605.2 | 16.66 ± 2.08 | 1.73 ± 0.22 |

Values correspond to the arithmetic mean and standard deviation of triplicate estimates.

## Methanogenic community structure and composition

To further explore differences in community structure and composition (β-diversity) between environmental microbial mat samples and communities following substrate incubation, Bray-Curtis distances were visualized through non-metric multidimensional scaling (NMDS) plots. Fig 2 indicates the dissimilarity between samples as ordinated by the metaMDS function with $k = 3$ dimensions and *stress* equal to *0.0386* for A4N5 and *0.145* for A5. The distribution of *mcrA*-based distances confirmed that environmental samples of both sites were clustered separately from those of the substrate-enriched treatments. Contrary, most of the treatment replicates did not show a clear clustering pattern across the different substrate incubations.

Particularly, in A4N5 (Fig 2A), the $H_2+CO_2+TMA$ enriched treatment exhibited the only grouping pattern among its replicates, suggesting a similar impact on the methanogenic community composition and, therefore, reproducibility in their incubation conditions. Whereas in A5, the $H_2+CO_2+TMA$ treatment (Fig 2B) showed the least overlap, suggesting high individual variability per replicate in the microbial community profiles compared to other treatments. The greater proximity observed between replicates of control and substrate-enriched treatments in A5 site suggests that the diversity of phylotypes among these samples is more homogeneous compared to that observed in the A4N5 replicates.

For exploratory purposes, an additional assessment accounting the average values of the observed relative abundance through all the replicates of each treatment for each site, was performed. The results showed a clearer pattern of distribution, forming three well separated clusters (S2 Fig). Similar to the replicates analysis, the environmental samples (both sites) were separated from the rest of the treatments as single-clusters. In addition, a clear cluster between control and $H_2+CO_2$ treatment was observed as well as a third separated cluster between the samples from the TMA and $H_2+CO_2+TMA$ treatments.

Taxonomic profile of the environmental samples from both sites at the order level revealed that in the microbial mat of A4N5, nine groups of methanogenic archaea, among which representative members of all methanogenic pathways described so far, were recovered. In turn, from the environmental samples of A5 microbial mats, 7 different groups of methanogenic archaea were detected. The A4N5 and A5 environmental methanogenic communities included class I and II hydrogenotrophic methanogens (Methanobacteriales, Methanococcales, Methanomicrobiales), class II methylotrophic methanogens (Methanosarcinales), strictly acetic methanogens (Methanotrichales); three previously non reported groups of methyl-reducing methanogens (Methanomassiliicocales, Methanonatronarchaeales and Methanomethyliales), and even a group of methanotrophic archaea known as *Ca*. Methanophagales (ANME-1) (S3 Fig).

To assess overall differences of the methanogenic communities between environmental samples and the substrate-enriched treatments, relative abundances of taxa recovered were evaluated at order-level (Fig 3). Methylotrophic methanogenesis, evidenced by sequences belonging to Methanosarcinales, was found to be a ruling pathway in the studied hypersaline microbial mats. However, this was not the only methanogenic group favored by the incubation conditions. An increase in the relative abundances of Methanomassiliicoccales was noticed mainly in the control treatment of both sites in comparison with the environmental samples. Besides the coexistence of both types of methylotrophic methanogens (conventional and $H_2$-dependent) Methanosarcinales/Methanomassiliicocales in all microcosm experiments, the enrichment with $H_2+CO_2$ also allowed the detection of the hydrogenotrophic Methanomicrobiales order in the microbial mats of A5. Another remarkable aspect of this experiment was that after incubations, some uncultured/unassigned *mcrA* gene sequences were still detected.

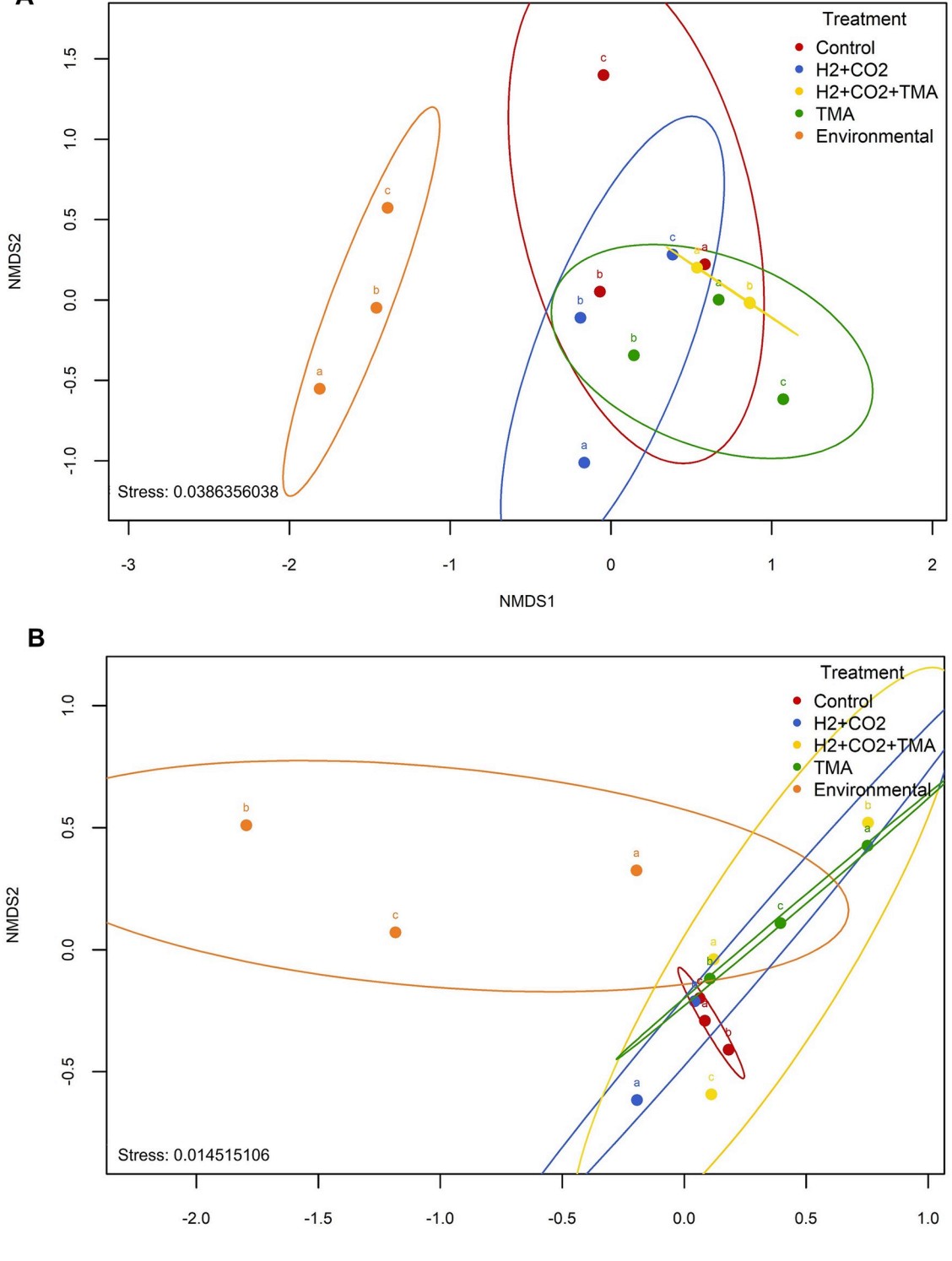

**Fig 2. NMDS ordination plot.** NMDS based on Bray-Curtis dissimilarity distance indices between environmental microbial mat methanogenic communities and communities following substrate-enriched incubations from A) A4N5 and B) A5. Greater distance between points indicates less similarity between their phylotypes. Stress values <0.2 mean a reliable representation of the data in two dimensions plot.

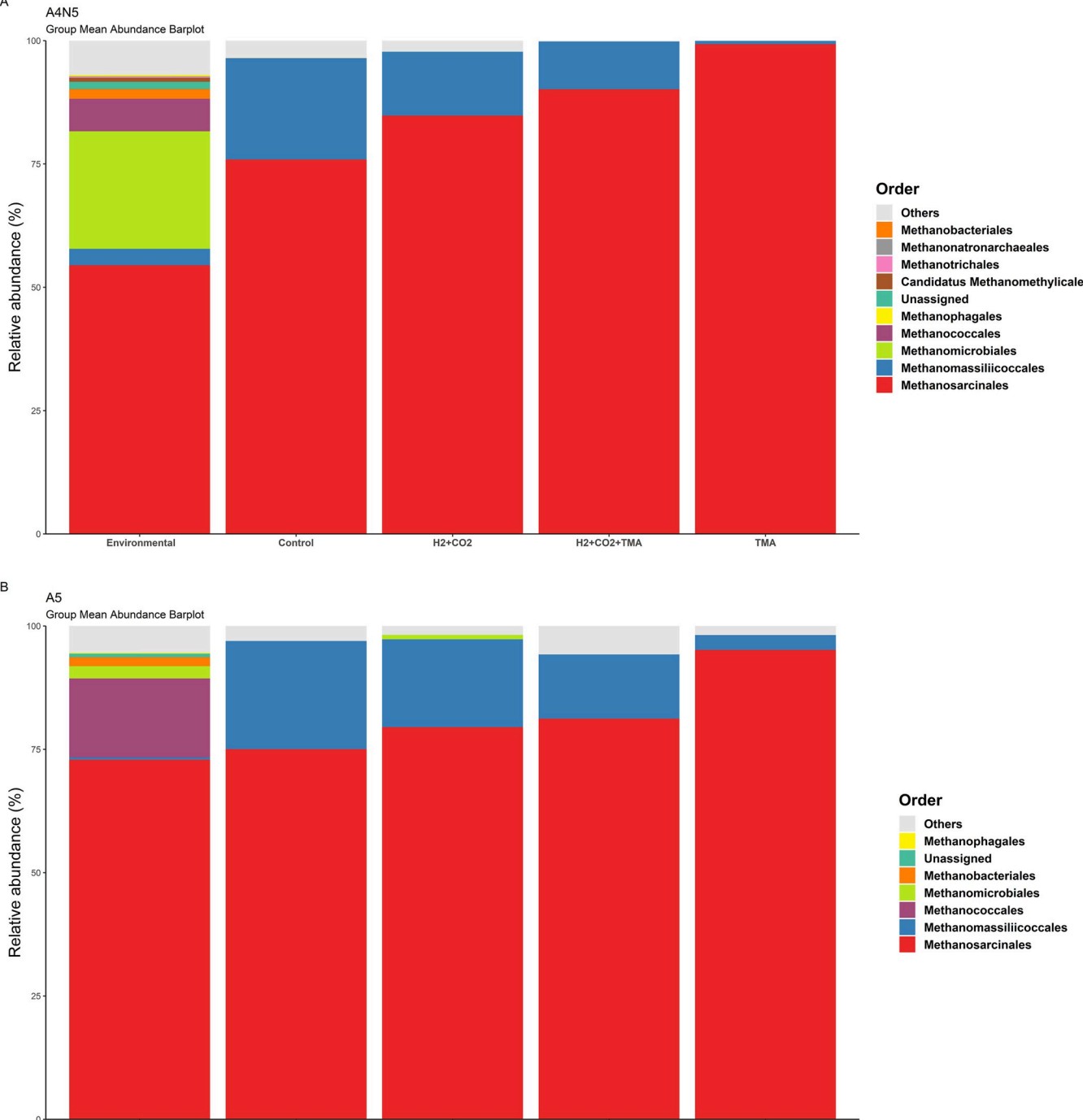

**Fig 3. Taxonomic relative abundance across samples.** Stacked bar plots depicting mean relative abundances of most abundant taxa in environmental (unmanipulated) microbial mat communities and communities following substrate-enriched incubations obtained from (A) A4N5 and (B) A5 at the Order level.

## Phylogeny of unassigned methanogenic OTUs

OTUs without taxonomic assignment (S2 Table) were mapped against a reference phylogenetic tree (Fig 4). Maximum likelihood analysis revealed a broad taxonomic distribution

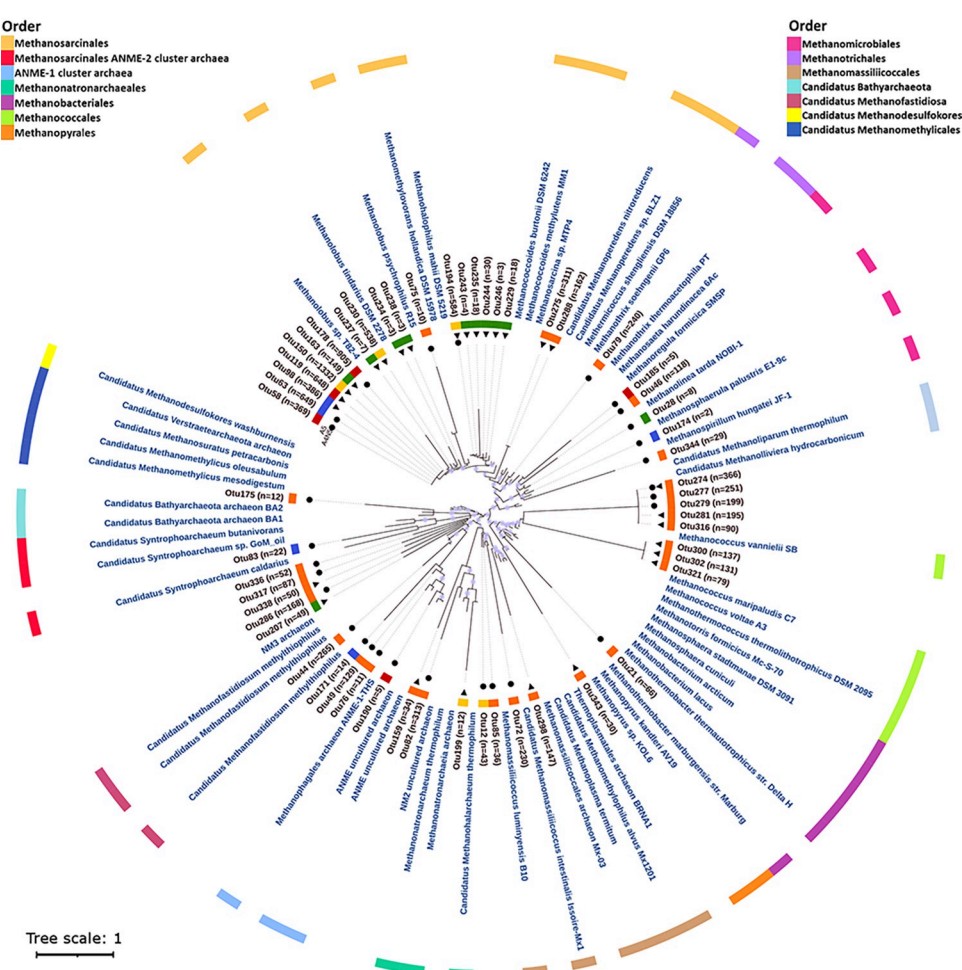

**Fig 4. Maximum likelihood phylogenetic tree.** Phylogenetic tree of uncultured/unassigned McrA amino acid sequences obtained in this study. The number of sequences for each OTU is shown in parentheses. Sampling sites are represented by circles (A4N5) and triangles (A5) at the end of branches. End nodes corresponding to methanogenic reference sequences are shown in blue. The sample type to which each OTU belongs is represented by the inner colored stripe (orange: environmental, red: control; blue: $H_2+CO_2$; yellow: $H_2+CO_2+TMA$ and green: TMA). Colorstrip in the outer ring corresponds to the reference methanogenic groups at the order taxonomic level.

spanning 3 distinct Archaea phyla: Ca. Bathyarchaeota, Ca. Thermoplasmatota and Euryarchaeota, mainly from OTUs recovered from environmental samples (inner orange label). OTU' sequences recovered from A4N5 microbial mats were placed within the Methanococcales clade (Otus 279, 277 and 274). Others were phylogenetically related to members of the orders Methanomicrobiales (Otus 46, 344), Methanobacteriales (Otu 21), Methanomassiliicoccales (Otu72), Methanofastidiosales (Otu 44), Methanothrichales (formerly Methanosaeta, Methanosarcinales, recently reclassified as Methanothrichales by the Genome Taxonomy Database, GTDB) (Otu 79), and Methanosarcinales (Otu 75) as well as a clade of archaea responsible for anaerobic methane oxidation, ANME-1 cluster archaea (Otus 76 and 49).

The phylogenetic estimation placed Otu 82 in a clade along a recently but partially described, methyl-reducing methanogen termed as NM2 [33], and Otu 175 as potentially related to members of the recently proposed phylum Bathyarchaeota. In turn, unassigned sequences recovered from environmental samples of A5 microbial mats showed phylogenetic relationships with methanogenic members of the orders Methanococcales (Otus 274, 321, 281,

300, 302, 316), Methanomassiliicoccales (Otu 298), and Methanosarcinales (Otus 288, 275), while the Otu 343 was found to be potentially related to members of the order Methanopyrales.

Also, the phylogenetic tree allowed us to estimate the potential phylogenetic relationship of uncultured/unassigned archaea OTUs recovered from the microcosm experiments. The phylogenetic estimation placed most of them within the Methanosarcinaceae family clade, of the order Methanosarcinales. Of these, three OTUs (Otus 119, 178 and 58) were recovered from the control experiments, nine (Otus 229, 246, 244, 235, 243, 238, 234, 237, 163) from the TMA incubations; while from the $H_2+CO_2$ and $H_2+CO_2+TMA$ incubation experiments two (Otu 98 and 63) and three OTUs (Otus 194, 230, and 150), were retrieved respectively. Regarding the OTUs, whose phylogenetic estimation placed them within other methanogenic orders, Otu 185 recovered from A4N5 control incubations (inner red label) suggested a potential relationship with *Methanoregula formicia* SMSP, of the order Methanomicrobiales, while the Otu 190 was placed within a cluster of methanotrophic ANME-1 archaea. In turn, unassigned methanogenic OTUs from $H_2+CO_2$ incubations (inner blue label) showed phylotypes related to members of orders Methanomicrobiales (Otu 174) and to ANME-2 cluster archaea (Otu 83). In addition to those related to Methanosarcinales, the sequences retrieved from the incubations with $H_2+CO_2+TMA$ (inner yellow label) were phylogenetically associated to methyl reducing methanogenic groups of the order Methanonatronarchaeales (Otu 199). Meanwhile from TMA incubations (inner green label), the Otu 28 showed a potential phylogenetic relationship to *Methanolinea tarda NOBI-1*, a member of the Methanomicrobiales order. The remaining OTU sequences (Otus 12, 49, 171, 336, 317, 338, 286, 207, 344) corresponded to lineages that were distantly related to previously known methanogenic/methanotrophic archaea and its phylogenetic estimation placed the most of them outside the Euryarchaeota clade.

## Discussion

Unlike most hypersaline aquatic systems in which methanogenesis has been studied, the Guerrero Negro microbial mats developed within brines of marine origin (thalassohaline) provide a unique opportunity to survey microbial communities by having a natural salinity gradient and different ionic composition than those of continental origin (athalassohaline) [34, 35]. This study aimed to highlight the presence and the importance of methanogenic archaea with high diversity but low-abundance, as well as an unknown physiology, present in hypersaline microbial mats, and the effect of the incubation conditions on diversity, structure, and composition of the sampled communities.

### Methanogenic diversity of samples in environmental condition

Consistent with prior research [10, 11, 13, 17], our findings agreed that methylotrophy was the dominant methanogenesis process, with the genera *Methanolobus* and *Methanohalophilus* as the main potential contributors to methane production at the studied sites (S1 Table). Cultured representatives of the slightly halophilic genus *Methanolobus* have been reported to utilize a broad range of substrates, including methylated sulfur compounds [36]. Such versatility may provide it with a competitive advantage over other methylotrophic groups and may explain the greater relative abundance of this group in the environmental samples at both sites. Similarly, *Methanohalophilus* genus comprises obligate halophilic methanogens that have only been reported to grow in methylated substrates. To balance the osmolarity of their cytosol, these microorganisms synthesize different organic solutes, such as betaine, which are precursors of TMA [37].

Regardless the dominance of *Methanolobus* and *Methanohalophilus*, microbial mats of A4N5 showed a higher methanogenic diversity compared to A5 (Table 2) and therefore a different methanogenic structure (Fig 2) and composition (Fig 3), despite the proximity of the sampling sites. Similar to methylotrophic methanogens, which use available osmolytes as substrate [38], *mcrA* gene sequences of potential methyl-reducers were recovered from A4N5 and A5 microbial mats. This confirms the simultaneous availability of $C_1$-methylated compounds and $H_2$/formate at sampling sites and the mechanisms of osmo-adaptation by these groups to the thalassohaline environment.

Along with uncultured members of Methanomassiliicoccales, sequences of *Ca. Methanohalarchaeum*, *Ca. Methanomethylicus* and *Ca. Methanosuratincola* (S1 Table), were recovered in A4N5 samples. These genera have not been previously identified in the Guerrero Negro microbial mats.

The ecophysiological description of the candidate genus *Methanohalarchaeum*, whose detection in A4N5 mats could be explained by its neutrophilic and extremely halophilic nature, is based on three highly enriched monomethanogenic cultures from sediments of hypersaline lakes in the Kulunda Steppe, Russia [39]. To the best of our knowledge, this is the first report of the genus in this particular site and under hypersaline, slightly alkaline environmental conditions.

Regarding the methyl-reducing *Ca. Methanomethylicus* and *Ca. Methanosuratincola* recovered in A4N5, metagenome-assembled genomes reconstruction has revealed their apparent preference for mesophilic habitats and potential ability to perform sugar fermentative metabolism via the Embden-Meyerhof pathway [40]. If so, this sugars fermentative metabolism capacity could serve as an additional source of acetate, which in these environments is commonly derived from a large source of sulfate-reducing bacteria that incompletely degrade organic compounds to acetate [41]. Therefore, the presence of acetoclastic methanogens, such as the *Methanosarcina* and *Methanothrix* genera (S1 Table), may be linked to the availability of acetate derived from these community members.

In addition, the presumed recognition of *Methanoculleus*, *Methanofollis*, *Methanotorris*, *Methanococcus*, *Methanocaldococcus* and *Methanobrevibacter* genera in A4N5, together with sequences taxonomically assigned to *Methanothermococcus*, *Methanobacterium* and *Methanolinea* in A5 (S1 Table), supports the idea that hydrogenotrophic methanogens are present in the extreme environmental conditions of Guerrero Negro [13, 16, 17].

Owed to little variation in pH and salinity between sampling sites A4N5 and A5 (Table 1), we attribute the lower richness of methanogenic genera observed in A5 (S1 Table) to distinct chemical trends exhibited by the evaporitic brines [42]. These trends could be affecting the availability of substrates for the different methanogenic metabolisms.

## Shifts in methanogenic communities under microcosm incubations

Rarefaction curves (S1 Fig) and the alpha diversity metrics (Table 2) revealed significant differences in overall microbial diversity between the samples of environmental and incubation conditions (Kruskal-Wallis, p = $1.44037e^{-2}$). These results suggested that strict anoxic conditions and the addition of methanogenic substrates led to the selection of unique phylotypes, and to a notorious decrease in the diversity (richness and evenness) of the microbial community. It is important to note that although the control experiments were intended as a baseline to assess the effect of the different substrates on the microbial communities, they allowed us to recognize the presence of native methanogenic archaea that were favored just by the displacement of the oxygen atmosphere.

In turn, the beta diversity analysis estimated from the average relative abundances calculated from the three replicates of each treatment, suggested that the differences of diversity

were owed to the control condition (without substrates or electron donors) as well as the TMA substrate. The clustering observed between control and $H_2+CO_2$ treatments in A4N5, suggested that $H_2+CO_2$ condition did not strongly impact the communities. Nonetheless, in the A5 $H_2+CO_2$ sample a notorious stimulation of Methanomicrobiales, known to present the hydrogenotrophic methanogenesis pathway, was recovered (S3 Fig). In addition, the clustering of samples from TMA and $H_2+CO_2+TMA$ treatments, suggested that the TMA substrate was the main condition that impacted the diversity observed between these two treatments.

Among the methanogens favored by the different incubation conditions, the order Methanosarcinales, whose metabolism is mostly methylotrophic and acetoclastic [43], was the most abundant in all samples; probably due to its versatility regarding carbon sources utilized. Previous studies have yielded similar results [13, 16, 17], although this is the first report on the composition of the methanogenic community at the A4N5 site using next-generation sequencing. Notably, sequences of methyl-reducing methanogens of the order Methanomassiliicoccales were recovered from both sampled sites (Fig 3). While the first isolation of members of this order was found in human feces [43], it also has been found thriving in sediments with intermediate sulfate concentrations [44] and has been reported to outcompete hydrogenotrophic methanogens in low hydrogen threshold conditions [45]. Our results showed that despite the high-sulfate concentrations in this environment and in our treatments (roughly, both at 73 mM), members of this order were recovered, which suggest a metabolic adaptation to higher sulfate concentrations in thalassohaline environments.

Although hydrogenotrophic genera were detected in environmental samples at both sites (S1 Table), the hydrogen and carbon dioxide incubations did not significantly enriched hydrogenotrophic methanogens other than Methanosarcinales and Methanomicrobiales (A5) (Fig 3). This suggests that even under these incubation conditions, other microbial groups, such as sulfate-reducing bacteria, may have consumed these common substrates and outcompeted methanogens [8, 41]. In addition, with exception of most of the methylotrophic methanogens of the Methanosarcinaceae family, the rest require external hydrogen sources, and hence, compete with hydrogenotrophic methanogens for this common substrate [45]. Since the hydrogenotrophic methanogenesis consumes four molecules of hydrogen to reduce carbon dioxide to methane [7, 45, 46], and the methanogenesis from methylated substrates require only one; the methyl-reducing methanogens may have an energetic advantage over the hydrogenotrophic methanogens at low hydrogen partial pressures [45]. In agreement with this hypothesis, the methyl-reducing methanogens found in our experiments, related to the order Methanomassiliicoccales, could have thrived due to a low hydrogen partial pressure in the $H_2 + CO_2$ and $H_2+CO_2+TMA$ experiments (Fig 3; S1 Table).

There's evidence that hydrogenotrophic methanogens increase their growth yield when the hydrogen concentration is low. This indicates that the energy conservation through the coupling between heterodisulfide (CoM-S-S-CoB) reduction and the ADP phosphorylation is stronger at lower hydrogen concentrations [47]. Thermodynamically, the free energy change $\Delta G°'$ from the $CO_2$ reduction to methane by hydrogenotrophic methanogens, diminished from -131 kJ mol$^{-1}$ at 105 Pa, to -30 kJ mol$^{-1}$ to the standard hydrogen concentrations in methanogenic ecosystems where the hydrogen partial pressure is around 1 to 10 Pa [47]. Alternatively, because the methanogen-growth varies depending on the type of metabolism [47, 48], where hydrogenotrophic ($\Delta G°'$, -131 kJ mol$^{-1}$ methane) and methyl-reducing pathways ($\Delta G°'$, -112.5 kJ mol$^{-1}$ methane) appear to have a higher net energy available compared to methylotrophic methanogenesis ($\Delta G°'$, -76 kJ mol$^{-1}$ methane) the low growth yields of methanogenic archaea other than Methanosarcinales, could be related to interspecies transfer of fermentation products ($H_2$), which leads the real thermodynamic values distant from standard conditions [47, 48].

Also, it is well known that thermodynamically, the sulfate-reducing bacteria have a greater affinity for hydrogen and acetate and can use them more efficiently [8, 41, 47]. Therefore, it was thought that the co-occurrence of methanogenesis and sulfate reduction in hypersaline environments was only due to the use of non-competitive substrates such as methanol, methane thiol, and methylamines [8]. However, recent studies [15, 16] have proposed that acetoclastic and hydrogenotrophic methanogens can coexist with SRB in environmental contexts similar to ours. The detection of sequences related to the acetoclastic genera *Methanosarcina* and *Methanothrix* in the A4N5 microbial mats, the detection of hydrogenotrophic orders in environmental samples from both sites, as well as Methanomicrobiales in the A5 $H_2+CO_2$ treatment, supports the idea of coexistence of these metabolisms even in hypersaline environments with high rates of sulfate reduction such as Guerrero Negro.

The greatest abundances of Methanomassiliicoccales members observed in incubations without methanogenic substrates (referred to as *control*) suggests that the replacement of the oxic atmosphere, rather than the addition of substrates, may be more conducive to methyl-reducing methanogenesis. Thus, the anoxic conditions may have favored fermentative processes and consequently the availability of hydrogen and $C_1$-methylated compounds for this specific methanogenic group [45, 49]. The anoxia and dark conditions used in the experiments were intended to emulate the deeper layers of microbial mats, where organic matter degradation processes have been shown to be more active [34]. However, we are aware of the limitation of using experimental approaches to evaluate biogeochemical processes in hypersaline environments. The given conditions allowed specific groups of methanogens to grew in enrichments and increased their relative abundance.

One term that contextualizes the significance of this study corresponds to "rare biosphere". This microbial ecological concept refers to ubiquitous highly diverse, low-abundance and mostly unknown microbes, that account for the majority of the observed phylogenetic diversity in a community and may significantly contribute to overall microbial activity [50, 51]. The data obtained from our microcosm experiments indicate that both hydrogenotrophic and methyl-reducing methanogenic archaea may be attributed to this group in hypersaline microbial mats of Guerrero Negro. These rare and low-abundance archaea may play a key role in the breakdown of organic matter and have the potential to become dominant under favorable conditions, thereby providing resiliency mechanisms for mediating ecosystem stability and function [51].

## Phylogeny of low-abundance unassigned methanogenic OTUs of Guerrero Negro B.C.S.

Since there is good correspondence between phylogeny and phenotype of methanogenic archaea [7], phylogenetic analysis of *mcrA* sequences is currently an important approach to explore the evolutionary divergence of uncultured and presumably novel methanogenic lineages. The current study, along with previous research [16] on these microbial mats have recovered environmental *mcrA* sequences that do not match any known methanogenic group. This reinforces the hypothesis of the presence of specific environmental clusters of methanogenic archaea in Guerrero Negro (Fig 4).

In addition to the taxonomically assigned groups (Fig 3), our results uncovered 55 low-abundance methanogenic OTUs phylogenetically distant from known genera (Fig 4; S2 Table), suggesting a higher diversity of methanogens in hypersaline environments with high sulfate concentration than previously reported [11, 13, 16, 17].

Most of the unassigned OTU sequences recovered from environmental samples were associated with hydrogenotrophic orders (A4N5: Methanococcales, Methanobacteriales,

Methanomicrobiales; A5: Methanococcales and Methanopyrales) and with the deep-branching archaeal lineage *Candidatus* phylum Bathyarchaeota (Fig 4). Since genomic evidence indicates that Bathyarchaeota members possess a wide range of metabolic capabilities beyond the methane metabolism, including acetogenesis, and dissimilatory nitrogen and sulfur reduction [52], the detection of related phylotypes encourages further exploration of the involvement of Bathyarchaeota in local and global biogeochemical cycles. Also, it has been reported that members of the Bathyarchaeota co-occur more frequently with hydrogenotrophic methanogenic orders than with other archaeal groups [53], reinforcing a possible syntrophic association between these archaeal groups.

With the evolutionary diversification of archaea lineages largely driven by the adaptation to environmental conditions and available carbon and energy sources [54], the identification of sequences related to hydrogenotrophic and methyl-reducing methanogens of the orders Methanonatronarchaeales, Methanomassiliicoccales and *Candidatus* Methanofastidiosa (Fig 4) is probably associated with an adaptive strategy of these groups to co-exist with conventional halophilic methylotrophic archaea that predominate in this environment.

Regarding the phylogenetic distribution of unassigned OTUs retrieved from the microcosm experiments (Fig 4), the anoxic incubation of the microbial mats (*control*) evidenced the presence of phylotypes related to members of the Methanosarcinales and Methanomicrobiales orders reported in previous studies [16, 17]. In eutrophic environments, these orders are often found together [55], apparently because they differ in substrate utilization [5]. The co-occurrence of sequences associated with members of Methanomicrobiales and Bathyarchaeota was also observed in $H_2+CO_2$ incubations. However, further genomic data is required to confirm this assumption.

In addition to methanogenic archaea, McrA sequences associated with ANME members of the family *Ca*. Methanoperedenaceae and to the order Methanophagales were detected at both sampling sites (Fig 3; S1 Table). Furthermore, the phylogenetic placement of environmental unassigned OTUs within anaerobic archaea of the ANME-1 and ANME-2 clusters (Fig 4) raises the possibility of anaerobic methanotrophy in the microbial mats. However, even the detection of these sequences does not prove active methane oxidation in the microbial mats, as there is geochemical evidence to the contrary [56, 57].

Collectively, our data suggest that microcosm experiments under anoxic conditions and selective substrate amendment provide methanogenic archaea (that were not or were hardly detectable in the environmental samples) with more energy and carbon sources to increase in relative abundances. This, in addition to demonstrating the presence of other than conventional methylotrophic methanogens in high-sulfate hypersaline environments, highlights the importance of considering the temporal or spatial availability of these substrates when assessing taxa from the rare biosphere, and their potential to become active and more dominant once a favorable environmental change occurs. However, further research is needed to understand the metabolism of the uncultured archaea observed, their evolutionary divergence, and their proposed methanogenic capacities.

## Conclusions

This study enhances the understanding of the diversity of *mcrA*-encoding archaea in hypersaline microbial mats and provides insights into the metabolic requirements of low-abundance methanogens in the Guerrero Negro microbial mats. Microcosm experiments evidenced the presence, co-occurrence and potential activity of different methanogenic archaea under anoxic and substrate addition conditions. The previously undescribed, low-abundance, and phylogenetically diverse archaea presented in this study, outline the importance of further research on

methane metabolism and its derivatives, as well as its significance in hypersaline environments.

## Supporting information

**S1 Fig. Rarefaction curves.** Rarefaction curves based on the arithmetic mean of OTUs observed in the microbial communities of both environmental samples and treatments (control and enrichments) from both sampling sites. The $x$ axis indicates the size of the library, and the $y$ axis represents the number of OTUs detected by each sample.
(PNG)

**S2 Fig. NMDS ordination plot of beta diversity using average values.** NMDS ordination based on Bray-Curtis dissimilarity distances accounting the average values of the observed relative abundance between replicates of environmental microbial mat methanogenic communities and replicates of each treatment for each site. Greater distances between points indicate less similarity between their phylotypes. Stress values <0.2 mean a reliable representation of the data in two dimensions plot.
(PNG)

**S3 Fig. Relative abundance at order level.** Bar chart of the relative abundance (%) of the methanogenic communities of microbial mats at the order level from A4N5 and A5 environmental samples and incubations (control and treatments).
(PNG)

**S1 Table. Relative abundances of methanogens across A4N5 and A5 environmental microbial mat samples at the genus level determined by high-throughput sequencing of *mcrA* gene fragments.**
(XLSX)

**S2 Table. Absolute frequency of recovered OTUs that were not taxonomically assigned to known methanogenic archaea.**
(XLSX)

## Acknowledgments

We are grateful to Exportadora de Sal, S.A. de C.V. for granting us access to the Guerrero Negro field sites. Additionally, we extend our thanks to Dr. Alejandra Escobar-Zepeda for her advice in the phylogenetic analysis and Abril Gamboa for lab assistance.

## Author Contributions

**Conceptualization:** Patricia J. Ramírez-Arenas, José Q. García-Maldonado, Alejandro López-Cortés.

**Data curation:** Patricia J. Ramírez-Arenas.

**Formal analysis:** Hever Latisnere-Barragán.

**Funding acquisition:** José Q. García-Maldonado, Alejandro López-Cortés.

**Investigation:** Patricia J. Ramírez-Arenas, Hever Latisnere-Barragán, José Q. García-Maldonado, Alejandro López-Cortés.

**Methodology:** Patricia J. Ramírez-Arenas, Hever Latisnere-Barragán.

**Project administration:** Alejandro López-Cortés.

**Supervision:** José Q. García-Maldonado, Alejandro López-Cortés.

**Visualization:** Hever Latisnere-Barragán, José Q. García-Maldonado.

**Writing – original draft:** Patricia J. Ramírez-Arenas.

**Writing – review & editing:** Hever Latisnere-Barragán, José Q. García-Maldonado, Alejandro López-Cortés.

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
