## [Decision Letter · Decision Letter 0]

17 Jun 2024

PONE-D-24-15528Highly diverse – low abundance methanogenic communities in hypersaline microbial mats of Guerrero Negro B.C.S., assessed through microcosm experimentsPLOS ONE

Dear Dr. García-Maldonado,

Thank you for submitting your manuscript to PLOS ONE. After careful consideration, we feel that it has merit but does not fully meet PLOS ONE’s publication criteria as it currently stands. Therefore, we invite you to submit a revised version of the manuscript that addresses the points raised during the review process.

We look forward to receiving your revised manuscript.

Kind regards,

Yizhi Sheng

Academic Editor

PLOS ONE

Journal Requirements:

   "This work was supported by Consejo Nacional de Ciencia y Tecnología (CONACYT) through grant FORDECYT-PRONACES, CF-2019-848287 to A.L.-C. and J.Q.G.-M."

5. We note that Figure 1 in your submission contain map/satellite images which may be copyrighted. All PLOS content is published under the Creative Commons Attribution License (CC BY 4.0), which means that the manuscript, images, and Supporting Information files will be freely available online, and any third party is permitted to access, download, copy, distribute, and use these materials in any way, even commercially, with proper attribution. For these reasons, we cannot publish previously copyrighted maps or satellite images created using proprietary data, such as Google software (Google Maps, Street View, and Earth). For more information, see our copyright guidelines: http://journals.plos.org/plosone/s/licenses-and-copyright.

Reviewers' comments:

Reviewer's Responses to Questions

**Comments to the Author**

1. Is the manuscript technically sound, and do the data support the conclusions?

Reviewer #1: Yes

Reviewer #2: Yes

2. Has the statistical analysis been performed appropriately and rigorously? 

Reviewer #1: Yes

Reviewer #2: Yes

3. Have the authors made all data underlying the findings in their manuscript fully available?

Reviewer #1: Yes

Reviewer #2: Yes

4. Is the manuscript presented in an intelligible fashion and written in standard English?

Reviewer #1: Yes

Reviewer #2: Yes

5. Review Comments to the Author

Reviewer #1: This manuscript, submitted by Patricia Ramírez-Arenas and colleagues, is both well-presented and very interesting. In this research, the authors sampled two microbial mats from a saltern located in Mexico with a salinity of about 8%. They conducted an exhaustive study of the taxonomic diversity of methanogenic archaea by high-throughput sequencing of the mcrA gene. The study also included microcosm environments with competitive (H₂/CO₂) and non-competitive (trimethylamine) substrates for sulfate-reducing bacteria.

I found the manuscript very well-written, carefully prepared, and clearly presented. The main original message is that, in a hypersaline environment, the predominance of methylotrophic methanogens is moderated by the presence of hydrogenotrophic and acetoclastic methanogens, confirming previous findings of coexisting methanogens and sulfate-reducing bacteria (SRB) competing for the same substrates. This finding is very relevant as hypersaline environments are a major global landscape, including saline marshes, coastal ecosystems, and salt flats and lakes, among others.

I have no major comments, but I would suggest including the sulfate concentration (if available) in Table 1. If the exact concentration is not available, an estimate (7-10% of total salts) would be helpful.

Reviewer #2: The manuscript titled “Highly diverse – low abundance methanogenic communities in hypersaline microbial mats of Guerrero Negro B.C.S., assessed through microcosm experiments” presents interesting and valuable findings on methanogenic communities in hypersaline environments. The manuscript is a valuable addition to the literature on microbial ecology and biogeochemistry. There are a few points raised below should be addressed to strengthen the study and its contributions to the field.

The statistical methods used to analyze the data need to be defined. Specify which statistical tests were performed, the rationale for their selection, and how the data met the assumptions of these tests. Include information on the software used for the analysis.

What is the logic to select methanogenic substrate combinations of H2/CO2 and N2 in the microcosm experiments? Why dimethylamine was selected specifically?

The discussion section should explore how the observed shifts in methanogenic community composition under different substrate conditions relate to broader ecological and biogeochemical processes in hypersaline environments.

High-resolution images should be provided for better clarity and readability. The current resolution of Figures 3 and 4 is not sufficient for readability and publication.

Check the citation format and make them consistent throughout. For example:

In the introduction section: “...More than half of global methane emissions are derived from microbial activity [3], and …”

However, in the discussion section: “...Since there is good correspondence between phylogeny and phenotype of methanogenic archaea (7),...”

6. PLOS authors have the option to publish the peer review history of their article (what does this mean?). If published, this will include your full peer review and any attached files.

Reviewer #1: No

Reviewer #2: No

---

## [Author Response · Author response to Decision Letter 0]

18 Jul 2024

We greatly appreciate your valuable comments, which were very important to improve the quality of our manuscript. Below you will find the answers to all the Editor and Reviewers comments. 

Editor’s comments.

Comment No 1: “Please ensure that your manuscript meets PLOS ONE's style requirements, including those for file naming…”

Response: We have made an additional revision of style requirements and modified the manuscript according to it. In particular, the nomenclature of the supplementary material has been changed, as well as the font size and style used in the sections, headings, and subheadings. 

Comment No 2: “In your Methods section, please provide additional information regarding the permits you obtained for the work. Please ensure you have included the full name of the authority that approved the field site access and, if no permits were required, a brief statement explaining why”.

Response: Governmental permits were not required for this work. However, we highlighted in lines 79-80 that Exportadora de Sal, S.A. de C.V. provided us permits for access and sampling collection.

L79-80: “…that kindly provided access to their facilities and allowed us the collection of samples.”

Comment No 3: “Please state what role the funders took in the study. If the funders had no role, please state: "The funders had no role in study design, data collection and analysis, decision to publish, or preparation of the manuscript." If this statement is not correct you must amend it as needed.”

Response: The funding institution, Consejo Nacional de Ciencia y Tecnología (CONACYT), had no role in study design, data collection and analysis, decision to publish, or preparation of the manuscript. 

Comment No 4: When completing the data availability statement of the submission form, you indicated that you will make your data available on acceptance. We strongly recommend all authors decide on a data sharing plan before acceptance, as the process can be lengthy and hold up publication timelines. Please note that, though access restrictions are acceptable now, your entire data will need to be made freely accessible if your manuscript is accepted for publication. This policy applies to all data except where public deposition would breach compliance with the protocol approved by your research ethics board. If you are unable to adhere to our open data policy, please kindly revise your statement to explain your reasoning and we will seek the editor's input on an exemption. Please be assured that, once you have provided your new statement, the assessment of your exemption will not hold up the peer review process.

Response: Sequencing data files are now available and freely accessible at NCBI.

Comment No 4: “We note that Figure 1 in your submission contains map/satellite images which may be copyrighted. All PLOS content is published under the Creative Commons Attribution License (CC BY 4.0), which means that the manuscript, images, and Supporting Information files will be freely available online, and any third party is permitted to access, download, copy, distribute, and use these materials in any way, even commercially, with proper attribution.”

Response: Thank you for the observation. We have added a new Figure 1 and provided the attribution to the map source in the corresponding figure legend. We had also provided information regarding the terms of use information for the map. 

L86-89: “Maps Source: INEGI, Digital topographic map, 2024, México (https://gaia.inegi.org.mx/mdm6). The original maps were modified to show the study sites. Map modifications comply with terms of use required by INEGI for free distribution (https://www.inegi.org.mx/contenidos/inegi/doc/terminos_info.pdf).”

Comment No 5: Please review your reference list to ensure that it is complete and correct. If you have cited papers that have been retracted, please include the rationale for doing so in the manuscript text or remove these references and replace them with relevant current references. Any changes to the reference list should be mentioned in the rebuttal letter that accompanies your revised manuscript. If you need to cite a retracted article, indicate the article’s retracted status in the References list and also include a citation and full reference for the retraction notice.

Response: A revision was made; the reference list is complete, and no retracted papers have been cited. 

Answers to reviewer 1 comments and revision clarifications

We highly appreciate the reviewer's positive comments. The revision has been accordingly made.

Comment No 1: I would suggest including the sulfate concentration (if available) in Table 1. If the exact concentration is not available, an estimate (7-10% of total salts) would be helpful.”

Response: A new sentence regarding sulfate concentrations was included in lines 97-99. 

L97-99: “The salinity and sulfate concentrations were settled according to a chemical analysis performed on samples of natural brines during a 2019 fieldwork (74.34mM)”.

Answers to reviewer 2 comments and revision clarifications

We highly appreciate the reviewers’ insightful and helpful comments on our manuscript. The revision has been made according to the specific comments.

Comment No 1: The statistical methods used to analyze the data need to be defined. Specify which statistical tests were performed, the rationale for their selection, and how the data met the assumptions of these tests. Include information on the software used for the analysis.

Response: The requested information was included in the Materials and Methods section. Lines 174-195 were modified to provide detailed methodological information. Lines 183-191 described the software and statistical methods used to analyze the data.

L174-195: “...Operational Taxonomic Units (OTUs) were generated based on sequence similarity (97 % identity) and alignment BLASTx score [20]. Chimeric sequences were removed employing Usearch v. 10.0.240 [21] and sequences with no hits were discarded from further analysis.

Taxonomic labels of OTU representative sequences were retrieved from the BLASTx best hit against the UniProt database described. Taxonomic rank names were obtained using the NCBI taxonomy database via TaxonKit v.0.10.1 [22]. 

Diversity and statistical analysis

The OTUs abundance table was exported to the R environment [23] for statistical analysis and visualization using the microeco package v1.7.1 [24] and the vegan [25], phyloseq [26], and ggplot2 [27] libraries. To assess differences in alpha diversity among the sampling sites and between environmental replicates and methanogenic substrate incubations, mean values were used for statistical analysis using the nonparametric Kruskal-Wallis test. The similarity patterns of the beta diversity of microbial communities were analyzed using a non-metric multidimensional scaling (NMDS) approach, with Bray-Curtis distance matrices employed as the basis for the analysis.

Phylogenetic analysis

OTU sequences without taxonomic assignment were translated into their corresponding Mcra peptide sequences via Once the identity of the representative sequences was resolved, EMBOSS Transeq [28] tool…”

Comment No 2: What is the logic to select methanogenic substrate combinations of H2/CO2 and N2 in the microcosm experiments? Why dimethylamine was selected specifically?

Response: The original text was revised and the rationale behind substrate selection has been addressed in lines 106-110. The objective was to stimulate the activity of low-abundance uncharacterized methanogenic archaea, which we hypothesized could be dormant in hypersaline environments due to a lack of sufficient substrate. It should be noted that the methylated compound used in this experiment was trimethylamine and not dimethylamine.

L106-110: “The selected methanogenic substrates were intended to stimulate ubiquitous, highly diverse, low-abundance microbes, with potential hydrogenotrophic (H₂+CO₂) and methyl reductive (H₂+CO₂+TMA) methanogenesis metabolisms. In turn, the use of TMA served as a positive control for methylotrophic methanogenesis, with N₂ employed to replace the oxic atmosphere in the microcosm experiments”.

Comment No 3: The discussion section should explore how the observed shifts in methanogenic community composition under different substrate conditions relate to broader ecological and biogeochemical processes in hypersaline environments.

Response: In order to relate how the observed microbial community shifts after the substrate incubation may be helpful for understanding ecological processes in hypersaline environments, lines 463-477 and 526-530 were incorporated into the Discussion section. 

L463-477: “The anoxia and dark conditions used in the experiments were intended to emulate the deeper layers of microbial mats, where organic matter degradation processes have been shown to be more active [34]. However, we are aware of the limitation of using experimental approaches to evaluate biogeochemical processes in hypersaline environments. The given conditions allowed specific groups of methanogens to grow in enrichments and increased their relative abundance.

One term that contextualizes the significance of this study corresponds to "rare biosphere". This microbial ecological concept refers to ubiquitous highly diverse, low-abundance and mostly unknown microbes that account for the majority of the observed phylogenetic diversity in a community and may significantly contribute to overall microbial activity [50,51]. The data obtained from our microcosm experiments indicate that both hydrogenotrophic and methyl-reducing methanogenic archaea may be attributed to this group in hypersaline microbial mats of Guerrero Negro. These rare and low-abundance archaea may play a key role in the breakdown of organic matter and have the potential to become dominant under favorable conditions, thereby providing resiliency mechanisms for mediating ecosystem stability and function [51]”.

L526-530: “This, in addition to demonstrating the presence of other than conventional methylotrophic methanogens in high-sulfate hypersaline environments, highlights the importance of considering the temporal or spatial availability of these substrates when assessing taxa from the rare biosphere, and their potential to become active and more dominant once a favorable environmental change occurs.”.

Comment No 4: High-resolution images should be provided for better clarity and readability. The current resolution of Figures 3 and 4 is not sufficient for readability and publication.

Response: Thank you for the observation. As recommended, a revision of the quality of the figures was made, and the figure files have been updated to meet the size and resolution criteria. In addition, we would like to clarify that the “uploading files platform” requires that files to be assessed previously to be uploaded and perhaps, the diminishing in resolution could be owed to this previous assessment that modified the original file. Individual figure files, when viewed separately, are of the requisite quality.

Comment No 5: Check the citation format and make them consistent throughout. For example: In the introduction section: “...More than half of global methane emissions are derived from microbial activity [3], and …” However, in the discussion section: “...Since there is good correspondence between phylogeny and phenotype of methanogenic archaea (7)...””

Response: Thank you for the observation. A detailed revision and corrections have been made accordingly.

---

## [Editor Report · Decision Letter 1]

26 Jul 2024

Highly diverse – low abundance methanogenic communities in hypersaline microbial mats of Guerrero Negro B.C.S., assessed through microcosm experiments

PONE-D-24-15528R1

Dear Dr. García-Maldonado,

We’re pleased to inform you that your manuscript has been judged scientifically suitable for publication and will be formally accepted for publication once it meets all outstanding technical requirements.

Kind regards,

Yizhi Sheng

Academic Editor

PLOS ONE
---

## [Editor Report · Acceptance letter]

5 Aug 2024

PONE-D-24-15528R1 

PLOS ONE

Dear Dr. García-Maldonado, 

I'm pleased to inform you that your manuscript has been deemed suitable for publication in PLOS ONE. Congratulations! Your manuscript is now being handed over to our production team.

Kind regards, 

on behalf of

Dr. Yizhi Sheng 

Academic Editor

PLOS ONE